# Microporous Adsorbents for CH_4_ Capture and Separation from Coalbed Methane with Low CH_4_ Concentration: Review

**DOI:** 10.3390/nano15030208

**Published:** 2025-01-28

**Authors:** Xiao Wei, Yingkai Xia, Shuang Wei, Yuehui Chen, Shaobin Yang

**Affiliations:** 1College of Material Science and Engineering, Liaoning Technical University, Fuxin 123000, China; weixiao19980412@163.com (X.W.); xiayingkai200719@126.com (Y.X.); weishuangcoyi@163.com (S.W.); 2School of Mining, Liaoning Technical University, Fuxin 123000, China; chenyuehui@lntu.edu.cn

**Keywords:** coalbed methane, adsorption, CH_4_/N_2_ separation, microporous adsorbents, progress

## Abstract

A rapid increase in natural gas consumption has resulted in a shortage of conventional natural gas resources, while an increasing concentration of CH_4_ in the atmosphere has intensified the greenhouse effect. The exploration and utilization of coalbed methane (CBM) resources not only has the potential to fill the gap in natural gas supply and promote the development of green energy, but could also reduce CH_4_ emissions into the atmosphere and alleviate global warming. However, the efficient separation of CH_4_ and N_2_ has become a significant challenge in the utilization of CBM, which has attracted significant attention from researchers in recent years. The development of efficient CH_4_/N_2_ separation technologies is crucial for enhancing the exploitation and utilization of low-concentration CBM and is of great significance for sustainable development. In this paper, we provide an overview of the current methods for CH_4_/N_2_ separation, summarizing their respective advantages and limitations. Subsequently, we focus on reviewing research advancements in adsorbents for CH_4_/N_2_ separation, including zeolites, metal–organic frameworks (MOFs), and porous carbon materials. We also analyze the relationship between the pore structure and surface properties of these adsorbents and their adsorption separation performances, and summarize the challenges and difficulties that different types of adsorbents face in their future development. In addition, we also highlight that matching the properties of adsorbents and adsorbates, controlling pore structures, and tuning surface properties on an atomic scale will significantly increase the potential of adsorbents for CH_4_ capture and separation from CBM.

## 1. Introduction

As we all know, with our global energy structure transforming for low-carbonization, traditional fossil fuels, such as coal and petroleum, are gradually decreasing in their share of the energy we consumption, while natural gas, as a relatively clean fossil fuel, has garnered increasing attention. In recent years, natural gas consumption has consistently shown an upward trend, and it exceeded 4000 bcm in 2019 [1]. However, the rapid growth in natural gas consumption poses a challenge, as conventional natural gas resources may be insufficient to meet demand in the future. The exploration and utilization of unconventional natural gas resources, such as coalbed methane (CBM), can offer a viable solution to filling this supply gap and promoting the development of clean energy. A significant category of unconventional natural gas, CBM primarily consists of methane, nitrogen, and minor amounts of carbon dioxide. According to the International Energy Agency, global CBM reserves exceed 260 tcm [2], highlighting its abundant availability. However, the concentration of CH_4_ in CBM is typically low (below 30%), and because of substantial impurities, CBM is usually directly emitted into the atmosphere rather than being collected and utilized, resulting in significant wastage of natural gas resources. Furthermore, CH_4_ is a potent greenhouse gas with a greenhouse effect approximately 21 times that of CO_2_ [3]. From 2002 to 2019, the CH_4_ concentration in the atmosphere rose from about 1700 ppbv to 1800 ppbv [4], and it had been projected to increase at an accelerated rate in the future, significantly exacerbating global warming. Clearly, the rational development and utilization of CBM enable the full exploitation of natural gas resources and facilitate the structural transformation of global energy in a low-carbon and green direction. More importantly, it can also help mitigate global warming and make significant contributions to environment-friendly and sustainable development. Currently, the efficient separation of CH_4_ and N_2_ mixtures represents one of the major challenges hindering CBM development. This paper provides an overview of technologies for CH_4_/N_2_ separation and the research progress in porous adsorbents for CH_4_ capture and CH_4_/N_2_ separation. We also delve into the impact of adsorbents’ structural and chemical properties on CH_4_ adsorption capacity and CH_4_/N_2_ separation selectivity, and discuss the challenges and opportunities associated with different adsorbents.

## 2. The Methods of CH_4_/N_2_ Separation

CH_4_ separation from coalbed methane is usually difficult. In addition to CH_4_ as the main component, coalbed methane also contains N_2_, CO_2_ and other light hydrocarbon compounds. In order to effectively transport, store, and use CH_4_, these impure gasses need to be removed. At present, these impurities, except N_2_, can be separated simply and efficiently by adsorption, absorption, cryogenic distillation, membrane separation, and hydrate methods. However, it is hard to separate CH_4_ and N_2_ because of their approximative physical and chemical properties. As shown in Table 1, the molecular kinetic diameters of CH_4_ (0.380 nm) and N_2_ (0.364 nm) are very similar, with a difference of only 0.016 nm. In addition, CH_4_ and N_2_ are both non-polar molecules and neither have dipole moments, while CH_4_ has a higher polarizability (26.6 × 10^−25^ cm^3^ for CH_4_, 17.6 × 10^−25^ cm^3^ for N_2_). Therefore, removing N_2_ from coalbed methane with low-concentration CH_4_ has greatly restricted the application of CH_4_.

The development of efficient N_2_ removal technology is very important for the separation and purification of CH_4_. At present, several methods have been applied to separate CH_4_ and N_2_, including cryogenic distillation, membrane separation, hydrate technology, and adsorption. Some of these methods have been successfully put to industrial-scale use, but there is still room for their improvement. In fact, the ideal technology for CH_4_ and N_2_ separation should have the characteristics of high efficiency, low cost, low energy consumption, high product purity, high yield, and excellent recovery rate. From this perspective, the development of separation and purification technology is moving in the direction of improving sufficiency, reducing energy consumption, and environmental friendliness.

### 2.1. Cryogenic Distillation

Cryogenic distillation is a common gas separation technology with high technical maturity, which has been widely applied on a large scale in oxygen production from air separation and other processes. In cryogenic distillation, the separation of gas mixtures is achieved by the difference in their liquefaction temperatures. Usually, cryogenic distillation uses mechanical methods such as throttling expansion or adiabatic expansion to compress gasses to induce the phase change, and then distill them using the difference in their boiling points. The gas with a low boiling point is retained in the liquid phase, while the gas with a high boiling point is enriched in the gas phase. Although the principle of cryogenic distillation for CH_4_/N_2_ separation is sample, and high-purity CH_4_ can be obtained with an excellent recovery rate, the disadvantages of this method are also obvious. Firstly, in order to obtain high-purity CH_4_, the liquefaction process needs to be carried out at extremely low temperatures (below −100 °C) and high operating pressures (about 13–25 atm), which requires a large amount of energy to achieve specific separation conditions, and the energy utilization efficiency is extremely low [5]. In addition, depending on the selected process and other impurities that may be present in feed gasses, the tail gasses may contain a large amount of CH_4_. Usually, the higher the CH_4_ recovery rate, the more energy is required to compress the gas mixture, which results in reduced energy efficiency when producing high-purity CH_4_. Furthermore, the equipment investment and operating costs of this method are extremely high. Cryogenic distillation is only suitable for large-scale methene separation industries. For small-scale coalbed methane, the cryogenic distillation process cost is even higher than that of producing liquefied natural gasses, and has no economic value [6]. In addition, the process parameters of specific cryogenic distillation can only reach the maximum separation efficiency when the gas mixture is treated with a specific CH_4_ concentration. There is no method based on overall planning to the separation of gas mixtures with large differences in CH_4_ content. At present, identifying methods of improving energy efficiency and reducing tail gas losses through system integration and process optimization is the main challenge for the development of cryogenic distillation in the future.

### 2.2. Membrane Separation Method

The membrane separation method is an efficient technology for gas mixture separation. The principle of the membrane separation method uses the gradient gas partial pressure of different membrane sides as the driving force, making use of the different permeabilities of gasses and selectivity differences in membranes to gas separation. Compared with traditional separation technologies, the membrane separation method has the advantages of having a low cost, flexibility, scalability, and having a small footprint [7]. In recent years, membrane technology has received increasing attention in CH_4_/N_2_ separation.

The core of the membrane separation method is membrane materials. An ideal membrane usually requires good structural stability, highly uniform pore structure, long service life, excellent separation selectivity, and great permeability. Two important indicators for evaluating membrane performance are permeability and selectivity. According to different separation mechanisms, membranes can be divided into CH_4_ selective membranes and N_2_ selective membranes. For CH_4_-selective membranes, the membrane permeability of CH_4_ is higher than that of N_2_, and product gasses with high CH_4_ concentrations are obtained on the low-pressure permeate side, which is convenient for pipeline transportation after pressurization. Meanwhile, for N_2_–selective membranes, CH_4_ is excluded from the retentate phase and left on the high-pressure side, which can effectively save the cost of gas recompression. Today, a variety of polymer membranes, inorganic membranes, and mixed-matrix membranes have been prepared for CH_4_/N_2_ separation [8,9,10,11,12]. In general, membranes of different types have different separation mechanisms. Polymer membranes usually separate CH_4_/N_2_ through differences in solubility and migration, while inorganic membranes produce a molecular sieving effect that is reflected by the modified size of gas channels. For polymer membranes, the similar physical and chemical characteristics between CH_4_ and N_2_ not only limit the efficiency of diffusion, but also trigger the reactions of diffusion and dissolution. The competitive relationship between separation selectivity and permeability makes it hard to separate CH_4_/N_2_, because if the selectivity is increased by 1 order of magnitude, the permeability will decrease by 4–5 orders of magnitude. In addition, it is very difficult to prepare inorganic membranes with both high permeability and excellent selectivity [13,14,15,16,17]. Although the shortcomings of membrane performances can be compensated by using muti-stage separation methods and multiple cycle processes in a CH_4_/N_2_ separation system, the addition of these processes will undoubtedly increase the separation cost and reduce the overall separation efficiency.

In summary, due to the limitation of membrane performances, the membrane separation method is not widely used in CH_4_/N_2_ separation. The application of membranes is still restricted by difficulties in preparation, poor stability, contradictions between selectivity and permeability, and an inability to process large-scale gas mixtures, and industrial application has not been achieved.

### 2.3. Hydrates Method

Hydrates are crystalline solid substances formed by gas molecules and water under specific conditions [18]. At specific temperatures and pressures, water molecules combine with each other through hydrogen bonds and form a cage-like structure. Gas and water molecules are combined through van der Waals forces to form hydrates. The gas separation technology, based on the hydrate method, uses the difference in the equilibrium pressure of hydrates formed by different components. When the reaction of hydrate generation reaches equilibrium, the gas components of the hydrate phase are different from those of the initial phase, thereby achieving gas separation [19]. For CH_4_/N_2_ separation, since the phase equilibrium pressure of CH_4_ needed to form hydrates is lower than that of N_2_, CH_4_ is preferentially captured in the stable cage-like structure formed by water molecules. After most of the N_2_ is separated in the gas phase, the CH_4_ hydrates are dissociated by reducing the pressure to create a gas stream of high CH_4_ concentration. In recent years, the hydrate method has received more and more attention, and several breakthroughs have been made in the fields of natural gas storage and gas separation [20,21,22,23]. The main advantage of the hydrate method is an excellent CH_4_ adsorption capacity. Under STP conditions, the CH_4_ storage capacity of 1 m^3^ hydrates is nearly 160 m^3^ CH_4_ gas [24]. In addition, the only reactants of hydrate formation are water and gas molecules, which have relatively little impact on the environment. Moreover, hydrates can be decomposed by heating and reducing pressure, and water can continue to participate in the process cycle. Although the reaction of hydrates can occur at room temperature, stable hydrate formation needs ultra-high pressures, which results in significant energy consumption and investment costs. Therefore, a feasible direction in which to break through the energy consumption limit is to add catalysts to reduce the formation pressure of CH_4_ hydrates. Recent research has found that additives such as cyclopentane (CP), tetrahydrofuran (THF), tetra-n-butylammonium (TBAB), rhamnolipid (Rha), and alkyl polysaccharides can effectively reduce the formation pressure of CH_4_ hydrates [23,25,26,27,28,29,30]. In addition, improved processes, such as mixed systems with nucleating substances, multi-step continuous stratification, and CO_2_ injection to replace CH_4_ in hydrates, have been proposed to improve separation efficiency [27,28,31]. Combining excellent hydrate additives with improved processes can effectively enhance the performance of the hydrate method.

In summary, because of the characteristics of high efficiency, safety, and simple progress, the hydrate method has competitive advantages in the specific context of CH_4_/N_2_ separation. However, current research on the hydrate method is still in its infancy. It is not only difficult to synthesize hydrates efficiently, the thermodynamics and kinetic mechanism of hydrate formation are also unclear. The commercialization of the hydrate method for CH_4_/N_2_ separation still faces many challenges.

### 2.4. Adsorption Separation Method

Adsorption separation is a well-known method of gas separation and purification that has unique advantages. Due to its low energy consumption, low cost, high energy utilization rate, and flexible operation, the adsorption separation method has been widely used in the gas production industry. The most commonly used technology of adsorption and separation is pressure swing adsorption (PSA), which is based on the different adsorption behaviors of different gases on adsorbents at a certain pressure [32]. At high pressures, the component with the greater adsorption force is preferentially adsorbed on the adsorbents, while the component with a weak adsorption force is retained in the gas phase, separated and concentrated as the light component at the outlet of the adsorption column. After the light component is separated, the adsorbed gas is gradually desorbed by reducing the pressure and is collected as a heavy component. The essence of CH_4_/N_2_ separation by PSA is the result of the differential adsorption–desorption behaviors of CH_4_ and N_2_ on adsorbents, which is mainly based on two principles: the thermodynamic equilibrium effect and the kinetic effect [33]. The thermodynamic equilibrium effect is based on the difference in adsorption strength of CH_4_ and N_2_ on adsorbents. CH_4_ possesses a stronger adsorption strength than N_2_ and is preferentially adsorbed. When the thermodynamic equilibrium is reached, the adsorption capacity of CH_4_ is higher than that of N_2_. At this time, N_2_ is separated as the light component in a pressurized adsorption process, while CH_4_ is separated in the decompression process. The kinetic effect is based on the difference in the diffusion rate of CH_4_ and N_2_ on adsorbents. Because the molecular kinetic diameter of CH_4_ is larger than that of N_2_, the diffusion resistance of CH_4_ in the pores is greater than that of N_2_, which makes N_2_ diffuse preferentially on adsorbents, while CH_4_ is retained outside of the pores for a relatively short time. Unlike the thermodynamic equilibrium effect, when CH_4_/N_2_ separates by kinetic effects, N_2_ is preferentially diffused and adsorbed on adsorbents in the pressurization process, and a high-purity CH_4_ stream can be directly obtained at the outlet, reducing the energy consumption of pressurization. Therefore, the PSA based on kinetic effect is mainly used in the purification of natural gas with a high CH_4_ concentration (>70%), and is not suitable for CH_4_ separation from natural gas with a low CH_4_ concentration [34,35]. Compared with the cryogenic distillation method, although the CH_4_ recovery rate of cryogenic distillation is higher, the energy consumption of PSA is much lower, and PSA has obvious advantages in energy utilization and production costs. In addition, the scale of equipment needed for PSA is small, making PSA more suitable for small-scale CH_4_ separation. Another feature of PSA is its flexible operation, as the technical parameters can be adjusted according to different conditions, and it is appropriate to use in a variety of production conditions and indicators. In application, except for flexibly selecting PSA processes based on conditions and needs, more attention should be paid to the integration and optimization of processes and adsorbents with excellent performances, which is also an important factor in the development of PSA for CH_4_/N_2_ separation.

## 3. Adsorbents for CH_4_/N_2_ Separation

The adsorption separation of CH_4_ and N_2_ on adsorbents is mainly based on three different effects: the thermodynamic equilibrium effect, the kinetic effect, and the molecular sieving effect. The thermodynamic equilibrium effect takes advantage of the fact that the interaction between CH_4_ and the adsorbents is stronger than that between N_2_ and the adsorbents, because CH_4_ exhibits a higher polarizability, causing CH_4_ to be selectively adsorbed. The kinetic effect is based on the fact that the kinetic diameter of CH_4_ is larger than that of N_2_, and the diffusion rate of CH_4_ is lower than that of N_2_ in specific pores, where the pore size is within a certain range. N_2_ preferentially diffuses in pores, achieving separation via the kinetic effect. Similarly to the kinetic effect, the molecular sieving effect takes advantages of the difference in the kinetic diameters of CH_4_ and N_2_. However, the pore size of the adsorbents is smaller, reaching a size larger than N_2_ but smaller than CH_4_, which causes N_2_ to enter the pores while CH_4_ is blocked outside of the pores. Obviously, an excellent adsorbent acts as the core and foundation of CH_4_/N_2_ separation by adsorption technology, and the adsorption separation process should be systematically selected and optimized according to the separation principle of the adsorbents. Generally speaking, excellent adsorbents usually have high adsorption capacity and good selectivity. In addition, good mechanical and regenerative properties, easy production, and non-reactivity to adsorbates also determine whether adsorbents can be used for adsorption separation. At present, it is still a difficult task to find an ideal adsorbent for efficient CH_4_/N_2_ separation, because there is a serious conflict between the interaction of CH_4_ and N_2_ with adsorbents. The higher polarizability of CH_4_ determines the higher selectivity of CH_4_ on adsorbents, while the smaller kinetic diameter of N_2_ leads to a higher diffusion rate on adsorbents. That is, the thermodynamic equilibrium effect and kinetic effect of CH_4_/N_2_ separation are in conflict. The difficulty of CH_4_/N_2_ separation is significantly higher than CH_4_ separating from other gases such as CH_4_/CO_2_ because the synergistic effect between the thermodynamic equilibrium effect and the kinetic effect is conducive to CH_4_/CO_2_ separation [33,36,37]. At present, a variety of adsorbents have been developed for CH_4_/N_2_ separation, including zeolites, metal–organic frameworks (MOFs), and porous carbons.

### 3.1. Zeolites

Zeolite is a typical aluminosilicate with regular micropores, and its structure is usually tetrahedron, hexahedron, or octahedron and composed of silicon oxide. The specific pore structures are formed by combinations of shared vertices and surface edges, varying with changes in the crystal structure. Zeolites possess the characteristics of a rich grid, a regular spatial structure, abundant micropores, a large specific surface area, good ion exchange performance, controllable surface polarity, and stability, making them potential high-performance adsorbents [38,39]. Research on zeolites for CH_4_/N_2_ separation has been carried out previously, but it was found that the capacity for CH_4_ on traditional commercial zeolites is low, and they do not have the ability to induce efficient CH_4_/N_2_ separation [40,41,42]. Currently, many methods have been applied to improve the adsorption capacity of zeolites, such as pore structure optimization, ion exchange, molding process improvement, silicon–aluminum ratio adjustment, and compounding with other materials. Many zeolites with excellent properties have been prepared to separate CH_4_ and N_2_, as shown in Table 2.

Yaremov compared the adsorption performance of CH_4_ and N_2_ on silicate zeolites and phosphate zeolites and systematically analyzed the effects of porosity parameters and surface properties on gas adsorption. It was found that the micropore size imitation and the presence of cations and acidic proton sites both directly affect the adsorption selectivity, and the zeolite with an MFI channel structure had the best performance in terms of CH_4_/N_2_ separation [43]. Then, Silva measured the adsorption isotherms CH_4_ and N_2_ on a binder-free 5A zeolite. At 305 K and 3 bar, the adsorption capacities of CH_4_ and N_2_ were 1.60 mmol/g and 1.02 mmol/g, respectively, which were 20% higher than that on the commercial binder used: 5A zeolite [44]. Hao et al. discussed the effect of different concentrations of Na+ ion exchange on the separation performance of clinoptilolites, and found that the electrostatic field in pores could be accurately controlled by Na+ ion exchange. The clinoptilolite C-2 exchanged with 0.2 mol/L NaCl was best for kinetic CH_4_/N_2_ separation [34]. Then, Liu added coal tar to silicate-1 (Si/Al ratio over 400) and granulated it to synthesize a new type of zeolite-based composite material (Z/AC). The addition of coal tar allowed for it to be converted into porous carbon with a uniform pore structure, while the characteristics of zeolite were obtained, thus successfully avoiding the defects conferred by changing Si/Al ratio and breaking the porous structure during granulation. The CH_4_ adsorption capacity of Z/AC is 23.45 cm^3^/g, and CH_4_/N_2_ selectivity is higher than 4.0 at 298 K and 1 bar [45]. Subsequently, Wu synthesized an ammonium ion-exchanged Y-type zeolite. As shown in Figure 1, by simple ion exchange with tetramethylammonium cations (TMA^+^) and choline cations (Ch^+^), the energy distribution of CH_4_ during adsorption in TMAY and ChY zeolites significantly decreased, which greatly improved CH_4_/N_2_ separation performance. At 25 °C and 100 kPa, the CH_4_/N_2_ selectivity of TMAY and ChY reached 6.32 and 6.50, respectively [46]. After that, Li described four zeolite molecular sieves with electroneutral framework porous structures (UiO-7, AlPO4-33, AlPO4-17, and AlPO4-18) for CH_4_/N_2_ separation, as shown in Figure 2. The experimental results proved that UiO-7, with a topological ZON type, exhibited the highest CH_4_ adsorption capacity (0.82 mmol/g) and a selectivity of 4.4 for a CH_4_/N_2_ (CH_4_:N_2_ = 50:50) mixed gas. Their theoretical calculations also showed that when the pore size is close to 0.76 nm, the adsorption selectivity reaches the maximum value [47]. Studies have found that the eight-membered ring structure of the RHO zeolite is a potential site for CH_4_/N_2_ separation. Based on this, Wang and co-workers exchanged hydrothermally prepared RHO samples with alkali metal cations (Na^+^, K^+^ and Cs^+^) to regulate the pore size of the eight-membered ring. The results showed that K-RHO obtained by K^+^ ion exchange exhibited the best CH_4_/N_2_ separation performance. In the PSA simulation, the product processing capacity of K-RHO was 36% higher than that of the commercial zeolite adsorbent ETS-4 [48]. Sadeghi studied the CH_4_/N_2_ separation effect of TMA-Y zeolite prepared by exchanging Na-Y zeolite with tetramethylammonium chloride ions. It was found that at 303.15 K and 5.0 MPa, for a gas mixture of 10% CH_4_ and 90% N_2_, the CH_4_/N_2_ selectivity of TMA-Y was 4.7, which was much higher than that of Na-Y (2.2). Although the available adsorption volume decreased from 0.3 cm^3^/g to 0.17 cm^3^/g, the GCMC simulation results showed that the adsorption strength of CH_4_ on TMA-Y was much higher than that of Na-Y, causing a higher selectivity for TMA-Y [49]. In another study, Sharma used polyether polyol (PEPO) to modify the surface of NaY zeolite, and found not only that that the modified zeolite had a high specific surface area, but that the hydroxyl groups introduced by PEPO modification also improved the CH_4_/N_2_ selectivity. At 298 K and 5 bar, the maximum CH_4_/N_2_ selectivity was 3.95 [50]. Wei studied the effect of ammonium ion exchange on the CH_4_/N_2_ separation performance of sodium-mordenite (MOR) and found that with the increase in ammonium exchange, the CH_4_/N_2_ adsorption selectivity on NH4-MOR increased significantly, which was attributed to the increase in the difference in affinity of CH_4_ and N_2_ on NH_4_-MOR. Among them, NH_4_-MOR-95% showed a CH_4_/N_2_ selectivity up to 4.6, far exceeding the original Na-MOR (1.9) while maintaining a high CH_4_ adsorption capacity (16.8 cm^3^/g) at 25 °C and 1 bar [51].

**Table 2 nanomaterials-15-00208-t002:** Adsorption performances of zeolites for CH_4_/N_2._

Zeolites	CH_4_/N_2_ Selectivity	CH_4_ Adsorption Capacity (mmol/g)	Reference
Si-pentasil S-1, MFI	4.8 ^a^	1.3 ^a^	[43]
Ca A, LTA	1.8 ^a^	1.2 ^a^	[43]
Na Y	1.6 ^a^	1.9 ^a^	[43]
5A Zeolite	-	1.6 ^b^	[44]
Silicalite-1 powder	4.4 ^c^	0.83 ^c^	[45]
Z/AC-600	4.2 ^c^	1.05 ^c^	[45]
TMAY	6.32 ^c^	0.5 ^c^	[46]
ChY	6.5 ^c^	0.4 ^c^	[46]
AlPO_4_-33	3.8 ^d^	0.51 ^d^	[47]
UiO-7	4.4 ^d^	0.82 ^d^	[47]
2% PEPO-NaY	3.95 ^e^	0.92 ^e^	[50]
NH_4_-MOR	4.6 ^c^	0.75 ^c^	[51]
ZnY-almIM	4.52 ^c^	0.49 ^c^	[52]
ZK-5	4.2 ^c^	1.34 ^c^	[53]
Ag-ZK-5	11.8 ^c^	1.6 ^c^	[54]
IM-5@100	4.7 ^c^	0.79 ^c^	[55]

^a^ 253 K, 1 bar; ^b^ 305 K, 3 bar; ^c^ 298 K, 1 bar; ^d^ 298 K, 1 bar, breakthrough curve (CH_4_/N_2_ = 1/1); ^e^ 298 K, 5 bar.

Although the cation exchange can change the pore structure and surface properties of zeolites, organic ligands can also be introduced into zeolites by the metal cations contained within them, providing a potential way in which to improve the performance of CH_4_/N_2_ separation. Recently, Li reported a two-step modification method of introducing metal complexes into the pores of NaY zeolite by ion exchange (Zn^2+^) and ligand (almlM) grafting. Both the experimental and simulation results showed that specific pore size (0.4 nm) and surface functional groups improved separation efficiency. At 298 K and 1 bar, the CH_4_/N_2_ selectivity of ZnY-almlM reached 4.52, which was significantly higher than that of original NaY zeolite (1.52). The two-bed six-step PSA simulation also showed that the modified zeolite could enrich CH_4_ from 50% concentration to 90%, with a recovery rate of above 80% [52]. In addition, the smaller the crystal size of zeolites means that more adsorption sites can be exposed, and a higher diffusion rate of gases can be exhibited in zeolites, making them more conducive to the preferential adsorption of CH_4_. However, most zeolites studied so far are above the micrometer level, but synthesizing zeolites on a nanoscale is still a great challenge. In 2021, Yang successfully regulated β-cyclodextrin to synthesize the nanoscale zeolite ZK-5. The size of the zeolite ranged between 3 μm and 50–100 nm and showed high specific surface area (370 m^2^/g) and pore volume (0.22 cm^3^/g), and the CH_4_ adsorption capacity increased by about 64% (1.34 mmol/g). At 298 K and 1 bar, the CH_4_/N_2_ (CH_4_:N_2_ = 20:80) selectivity calculated by the IAST method reached 4.2. Kinetic experiments suggested that ZK-5 has higher diffusion and mass transfer rates than the zeolite on a microscale [53]. Subsequently, Kencana et al. prepared the silver ion-exchanged zeolite nanocrystal Ag-ZK-5 and reduced the crystal size to about 100 nm. Nanoscale Ag-ZK-5 possessed fast adsorption kinetics and good regeneration. The uniform distribution of Ag^+^ meant that Ag-ZK-5 exhibited excellent CH_4_/N_2_ separation ability. At 298 K and 1.0 bar, the CH_4_ adsorption amount reached 1.6 mmol/g, and the selectivity calculated by IAST was as high as 11.8 [54]. Tang obtained the nanoscale IMF zeolite IM-5@100 by simply adjusting the addition amount of aluminum source. As the addition amount of aluminum source decreased, the crystal size decreased significantly. Compared with the micron-sized IM-5@50, the mesopore volume of IM-5@100 increased from 0.12 cm^3^/g to 0.49 cm^3^/g, and the specific surface area increased from 38 m^2^/g to 118 m^2^/g, as shown in Figure 3. IM-5@100 exhibited a larger CH_4_ adsorption capacity (17.8 cm^3^/g) and a high selectivity (4.7) for CH_4_/N_2_ (CH_4_:N_2_ = 50:50), and had better mass transfer ability [55].

In recent years, with the rapid development of computer simulation technology, many methods, such as molecular dynamics (MD), grand canonical Monte Carlo (GCMC), and density functional theory (DFT), have been the focus of research in materials science, and possess unique advantages in exploring the adsorption behaviors of gases on materials with regular crystal structures and pores such as zeolites. Computer simulation technology can not only provide support for experimental results, but can also select potential materials with high CH_4_/N_2_ separation performance from a large number of candidates, providing guidance for the development of new zeolites. In 2017, Golchoobi used the GCMC method to analyze the adsorption of different gases on 13X zeolite. In his study, the structure of 13X zeolite was constructed with an FAU framework, and the interaction between gases and adsorbents was calculated by way of a general force field, and the adsorption isotherm was simulated using the metropolis method. The calculated isosteric heat of CH_4_ and N_2_ were 6.14 and 4.37 kcal/mol, respectively, and the simulated adsorption results were fitted to the Langmuir and Toth isotherm models [56]. After that, Fischer carried out GCMC simulation analysis on the adsorption performances of AlPO zeolite frameworks with 53 different structures for CH_4_ capture. The simulation results were compared with the experimental data to verify the force field parameters of CH_4_ adsorption on AlPO, and the framework with an ATV topology showed the best CH_4_/N_2_ separation selectivity. Although some topological AlPO zeolite molecular sieves do not exist, several potential high-performance zeolite adsorbents have been screened out [57]. In another report, the adsorption behaviors of CH_4_ and N_2_ on MER zeolite were simulated by GCMC and were in good agreement with the experimental results, proving the reliability of the simulation model and COMPASS force field. Based on GCMC simulation results, MD simulations were performed on the diffusion behavior of CH_4_ and N_2_. The results showed that the molecular configuration diffusion mode related to the pore size was dominant, and the cations outside the zeolite framework had an important influence on the diffusion of the gas, and the diffusion behavior was anisotropic [58].

In summary, with thorough research, a variety of zeolites with high adsorption capacity and separation selectivity have been developed for CH_4_/N_2_ separation, and are expected to be applied in industry for CH_4_ separation from natural gas. Recent researches have mainly focused on zeolites with a thermodynamic equilibrium effect for CH_4_/N_2_ separation, such as clinoptilolites, Y zeolites, and high-silicate zeolites obtained by ion exchange. In addition, the CH_4_/N_2_ separation performance is improved by composite synthesis and crystal size reduction. However, there are still many difficulties in developing zeolites with excellent performances for efficient CH_4_/N_2_ separation. For example, achieving low-cost, high-efficiency, and green zeolite preparation, solving the contradiction that CH_4_ is thermodynamically preferentially adsorbed and that N_2_ kinetically diffuses faster on zeolites, avoiding the reduction of CH_4_ adsorption capacity due to the preferential adsorption of polar molecules like H_2_O with strong polarization, and combining computer simulation methods with the development of new zeolites.

### 3.2. Metal–Organic Frameworks (MOFs)

Metal–organic frameworks (MOFs) are a kind of emerging crystalline material that are self-assembled by metal ions and organic ligands with coordination bonds. The most obvious feature of MOFs is their good designability. By adjusting the type of metal ions and the organic ligands, MOFs with different structures can be designed, thus different physical and chemical properties can be exhibited. The distinctive structures of MOFs give them the characteristics of large a specific surface area, high pore volume, and diverse pore structures, and MOFs have shown great application potential in the fields of optical catalysis, wastewater purification, energy storage, and gas separation [59,60]. Currently, MOFs have attracted more and more attention in the separation and purification of CH_4_. In this way, the unique microporous structure of MOFs is conducive to separating CH_4_ from other gas molecules with similar sizes. The large specific surface area provides many active sites for CH_4_ adsorption, and the controllable surface properties can further improve the efficiency of CH_4_ separation [61]. Recently, many MOFs have shown high CH_4_ adsorption capacity and high CH_4_/N_2_ separation selectivity, as shown in Table 3.

In 2010, Saha studied CH_4_/N_2_ separation performance of MOF-5 with zinc ion ligands, and found that the adsorption amount of CH_4_ at 298 K and 100 bar was 17.15 wt.%. Although MOF-5 has a good CH_4_ adsorption capacity, the CH_4_/N_2_ separation ratio is only 1.13, which does not induce the ability to separate CH_4_/N_2_ [62]. Later, Karra synthesized MOF-14 with a composition of Cu_3_(BTB)_2_(H_2_O)_3_(DMF)_9_(H_2_O)_2_. The chain-like structure of this material enhances the crystal stability and improves the adsorption capacity. In addition, the capacity of water adsorbed on MOF-14 is significantly lower than that of other MOFs with open metal sites, and the adsorption amounts of CH_4_ and N_2_ by MOF-14 at 298 K and 2 MPa are about 5 mol/kg and 2.5 mol/kg, respectively [63]. Li compared the performances kinetic separation of CO_2_/CH_4_ and CH_4_/N_2_ on MOF-5 and Cu_3_(BTC)_2_, and found that Cu_3_(BTC)_2_ with unsaturated metal sites formed by dehydration was more inclined to adsorb CO_2_ molecules, while MOF-5 with a more suitable pore size showed better ability to separate CH_4_/N_2_ [64]. In 2015, Sun’s team used volumetric chromatography (VC) and inverse gas chromatography (IGC) to measure the adsorption thermodynamics and kinetics of CH_4_ and N_2_ on Al-BDC for the first time, and the CH_4_/N_2_ selectivity of Al-BDC reached 4.3. The adsorption heats of CH_4_ and N_2_ at zero surface coverage measured by the IGC method were 15.3 kJ/mol and 11.5 kJ/mol, respectively, and the micropore diffusion rate of CN_4_ and N_2_ at 303 K were, respectively, 7.04 × 10^−8^ cm^2^/s and 1.58 × 10^−7^ cm^2^/s. The results show that the unique microporous structure of Al-BDC plays an important role in CH_4_ adsorption and CH_4_/N_2_ separation [65].

In fact, the unsaturated sites in MOFs have a great influence on gas adsorption, separation, and regeneration. Therefore, increasing the number of unsaturated sites for adsorption can enhance the multiple interactions between CH_4_ and MOFs, and enlarging the accessible surface area for adsorption can effectively increase the adsorption amount of CH_4_ on MOFs and improve the CH_4_/N_2_ separation selectivity. Based on this point, Li’s team designed a metal–organic framework M/DOBDC (M = Mg, Co, Ni) with a honeycomb-like structure and a high concentration of coordinated unsaturated sites, as shown in Figure 4. The cage pore diameter is about 12 Å, and the content of coordinated unsaturated sites is about 4.5 /nm^3^. Among them, Ni/DOBDC exhibits the best CH_4_/N_2_ adsorption selectivity (greater than 3) and the largest CH_4_ adsorption capacity (about 40 cm^3^/g). The excellent performances of Ni/DOBDC are attributed to the strong interaction between CH_4_ and coordinated unsaturated Ni ions [66]. Then, Hu further explored the effects of different metal ions on the adsorption performance of M_3_(HCOO)_6_ (M = Mg, Mn, Co and Ni), and found that the different affinity of metal ions for CH_4_ led to differences in CH_4_ adsorption capacity and CH_4_/N_2_ separation selectivity. In dynamic experiments, Ni_3_(HCOO)_6_ exhibited the largest CH_4_ adsorption capacity (1.09 mmol/g) and the highest CH_4_/N_2_ selectivity (6.5) at 0.4 MPa and 298 K, indicating that changes in metal ions play an important role in regulating pore size and surface properties [67]. In 2019, Chang reported a three-dimensional Cu-MOF with two microporous structures of different properties, which exhibited high CH_4_/N_2_ selectivity (10.00–12.67). At 298 K and 1.0 bar, the CH_4_ and N_2_ adsorption amounts were 14.17 cm^3^/g and 2.40 cm^3^/g, respectively. Density functional theory simulations showed that CH_4_ can form multiple van der Waals forces in both hydrophilic and hydrophobic pores, which improves the adsorption capacity of CH_4_. On the contrary, the adsorption amount of N_2_ was very low because of the weak polarity of hydrophobic pores and the occupation of open metal sites in hydrophilic pores by CH_4_ [68]. Then, Huang studied the adsorption and separation performances of CH_4_/N_2_ on two one-dimensional square Al-MOFs (CAU-10-H, MIL-160) and two rhombus counterparts (Al-Fum and MIL-53(Al)), as shown in Figure 5. Among them, Al-Fum showed extremely high CH_4_/N_2_ selectivity (17.2) and high CH_4_ adsorption capacity (1.14 mmol/g) at 273 K and 1.0 bar. The simulation calculation results showed that the synergistic effect of the strong affinity site (mu(2)-OH) in the structure and the polar pore surface greatly promote the adsorption of CH_4_. MOF development can take place by ensuring the exposure of active sites and the restriction effect generated by selecting suitable pores [69]. Chen and his colleagues reported a Zr-based metal–organic framework, MIP-203-F, whose one-dimensional microporous structure was divided into two symmetrical co-walled triangular pores by the -OH groups on side chains. This distinctive structure exposes a large number of synergistic polar sites, which is conducive to the adsorption of CH_4_ with high polarizability, and effectively overcomes the trade-off between the adsorption capacity of CH_4_ and CH_4_/N_2_ selectivity [70]. However, some researchers have also pointed out that the existence of open metal sites does not necessarily represent high CH_4_/N_2_ selectivity, and excessive interaction between adsorption sites and gas molecules may also be detrimental to the overall separation performance. Hu systematically studied the CH_4_ adsorption potential difference on four MOFs (Ni_3_(HCOO)_6_, Cu(INA)_2_, Al-BDC, and Ni-MOF-74) with similar grid topology. Among them, Ni-MOF-74 and Al-BDC are strong polar adsorbents with unsaturated metal sites, while Ni_3_(HCOO)_6_] and Cu(INA)_2_ lack polar groups in their frameworks and are considered to be non-polar (or weak polar) adsorbents. The adsorption potential trend of CH_4_ was Ni-MOF-74 > Ni_3_(HCOO)_6_ > Cu(INA)_2_ > Al-BDC, while the adsorption potential trend of N_2_ was Ni-MOF-74 > Al-BDC > Ni_3_(HCOO)_6_ > Cu(INA)_2_, and the CH_4_/N_2_ selectivity on Ni_3_(HCOO)_6_ and Cu(INA)_2_ was excellent (greater than 6). Only increasing the surface polarity cannot lead to effectively improving CH_4_/N_2_ selectivity, and the effect of pore size should also be considered [71]. Then, three MOFs (BUT-67, Zr-AbBA, and PCN-702) that can be used in CH_4_/N_2_ separation by a computer simulation method from more than 100 different Zr-MOFs were screened out. Their CH_4_/N_2_ selectivity was 4.5, 3.4, and 3.8, respectively, and the CH_4_ working capacities were, respectively, 3.6, 3.9, and 3.5 mol/kg. It is worth noting that these three Zr-MOFs do not contain open metal sites in their structures, but their separation performance is generally better than that of HKUST-1, which contains open metal sites [72]. Recently, Chang reported a new type of ultra-microporous copper-based MOF (NKMOF-8-Me) with non-polar pores and no open metal sites, and exhibited high structural stability and reproducibility. Not only was a high CH_4_ adsorption capacity (1.76 mmol/g) measured, but a high CH_4_/N_2_ breakthrough selectivity (7.8) was shown in the breakthrough experiment, and the isosteric heat (28.0 kJ/mol) was also higher than some MOF materials containing polar sites [73]. Therefore, the content of open sites is not the only factor that determines the adsorption performance. Factors including pore structure and functional groups will both affect adsorption behaviors, and need to be comprehensively considered when evaluating the overall adsorption and separation performance.

Since the kinetic diameters and thermodynamic properties of CH_4_ and N_2_ are very similar, ultra-micropore structures with specific sizes can efficiently separate CH_4_ and N_2_. Therefore, adjusting the pore structure can effectively improve the CH_4_/N_2_ separation selectivity. Kim proposed a simple method of improving the separation performances of Cu_3_BTC_2_. Utilizing the structural characteristics of hydrophilic and hydrophobic pores in the metal–organic framework of Cu_3_BTC_2_, H_2_O molecules were used to block the hydrophilic pores to prevent gas to enter, while the hydrophobic pores were used as gas adsorption sites. This novel method successfully increased CH_4_/N_2_ selectivity by six times to 24.7. However, the reduction in the adsorbable pore volume also resulted in a low CH_4_ adsorption amount, which was only 1.5 mmol/g at 100 bar and −30 °C [74]. Then, Chang reported a novel pore construction strategy using aliphatic ligands with appropriate sizes to construct specific cage structures for preferential CH_4_ adsorption, and a series of M-CDCs with different metal ion centers and trans-1,4-cyclohexanedicarboxylic acid (H_2_CDC) as ligands were synthesized, with a cage diameter of about 5.4 Å. These M-CDCs have better CH_4_/N_2_ selectivity (13.1–16.69), among which the adsorbent selection parameter of Al-CDC is as high as 82.0. In addition, M-CDCs demonstrate good regeneration performance, acid and alkali resistance, and heat resistance, and are expected to become a candidate adsorbent for CH_4_/N_2_ separation [75]. Columnar Ni-based MOFs (Ni(BTC)(BPY), Ni(BTC)(TED), and Ni(BTC)(PIZ)) were synthesized to improve the CH_4_/N_2_ separation selectivity by regulating the size of the interlayer channels. Among them, Ni(BTC)(PIZ), with a suitable pore size (0.611 nm) and abundant surface carboxylic acid groups, showed the best CH_4_ adsorption capacity (1.62 mmol/g) and CH_4_/N_2_ selectivity (7.32) [76]. Then, Wang prepared an MOF material Ni(ina)_2_ composed of Ni ions and isonicotinic acid (Figure 6), which had a specific ultra-micropore size (0.6 nm) and exhibited high CH_4_ adsorption (40.8 cm^3^/g) and CH_4_/N_2_ selectivity (15.8) under ambient conditions. Ni(ina)_2_ not only has good thermal and wet stability, but can also be scaled up at low cost, and has potential industrial application value [77]. After that, an aluminum-based metal–organic framework material, MIL-120Al, was reported, which formed non-polar pores with a pore size comparable to the kinetic diameter of CH_4_, achieving thermodynamic–kinetic synergistic separation of CH_4_/N_2_ mixtures. At 298 K and 1 bar, the adsorption of CH_4_ reached 33.7 cm^3^/g [78]. Later, Zhu’s team obtained a columnar multi-component MOF material [CuCe L(Cl-4-bdc)_0.5_(H_2_O)_2_·(H_2_O)_6_]_n_. Cu(II) and Ce(III) were used to construct 3d-4f layers as the substrate, and Cl4-bdc ligands were inserted between the layers as cushion layers, with an appropriate pore size and chlorine-modified channel surface. At 298 K and 1 bar, the CH_4_ adsorption capacity reached 28.41 cm^3^/cm^3^, while the N_2_ adsorption capacity was only 3.43 cm^3^/cm^3^, and the separation selectivity was 13.32 [79].

The particle size of MOFs is also an important factor for gas adsorption, and the small crystal size usually enhances the adsorption and separation performance of MOFs. In 2015, He reported a method of introducing flexible heterostructures into rigid MOF bulk phases by ligand exchange, which transformed the 3D rigid organic–metal framework into a 2D flexible network, and the crystal reversibly transformed from a non-porous structure to a porous structure, while the crystal size was significantly reduced. At 273 K and 10 bar, the CH_4_/N_2_ selectivity increased from 1.86 to 8.2, showing good potential for separating CH_4_/N_2_ [80]. Then, Wang compared the adsorption capacity of two 2D MOFs and 3D MOFs composed of Cu^2+^ and trifluoromethanesulfonate (OTf) (Figure 7), and found that 2D MOF showed better CH_4_/N_2_ selectivity. Although the pore volume and specific surface area were smaller, the layered structure exhibited a stronger adsorption potential, so the separation ability was better. In the breakthrough experiment, the breakthrough time difference between CH_4_ and N_2_ on 2D MOFs was 1.3 min, which was 30% higher than that of 3D MOFs [81]. Later, Feng used the characteristics of surface-induced low-energy-barrier-oriented crystallization to obtain a higher supersaturation at the solid–liquid interface, promote heterogeneous nucleation, and synthesize in situ two-dimensional Al-CDC nanosheets on the macroporous surface of polyacrylate (PA). The ultra-thin structure gives the composite higher adsorption efficiency and lower diffusion resistance. Compared with bulk Al-CDC, the adsorption efficiency increased by 1.73 times, and the CH_4_/N_2_ (CH_4_:N_2_ = 50:50) selectivity was as high as 13.75 [82].

**Table 3 nanomaterials-15-00208-t003:** Adsorption performances of MOFs for CH_4_/N_2._

MOFs	CH_4_/N_2_ Selectivity	CH_4_ Adsorption Capacity (mmol/g)	Reference
MOF-5	1.13 ^a^	1.07 ^a^	[62]
MOF-14	-	5.0 ^b^	[63]
MOF-5 particles	-	0.62 ^c^	[64]
Cu_3_(BTC)_2_ particles	-	0.91 ^c^	[64]
Al-BDC	4.3 ^d^	-	[65]
Ni/DOBDC	3 ^e^	1.79 ^e^	[66]
Ni3(HCOO)6	6.5 ^f^	1.09 ^f^	[67]
Cu-MOF	11 ^e^	0.63 ^e^	[68]
Al-Fum	17.2 ^g^	1.14 ^g^	[69]
Ni_3_(HCOO)_6_	6.2 ^e^	0.8 ^e^	[71]
Cu(INA)_2_	6.9 ^e^	0.8 ^e^	[71]
Al-BDC	3.0 ^e^	0.75 ^e^	[71]
Ni-MOF-74	1.4 ^e^	2.6 ^e^	[71]
Cu_3_BTC_2_	24.7 ^h^	1.5 ^h^	[74]
Ni(BTC)(PIZ)	7.32 ^e^	1.62 ^e^	[76]
Ni(ina)_2_	15.8 ^e^	1.82 ^e^	[77]
MIL-120Al	-	1.5 ^e^	[78]
NKMOF-8-Br	8.9 ^e^	1.84 ^e^	[83]
DMOF-A2	7.2 ^e^	1.65 ^e^	[84]

^a^ 298 K, 100 bar; ^b^ 298 K, 20 bar; ^c^ 298 K, 1 bar, breakthrough curve (CH_4_/N_2_ = 55/45); ^d^ 298 K, 1 bar, breakthrough curve (CH_4_/N_2_ = 50/50); ^e^ 298 K, 1 bar; ^f^ 298 K, 4 bar; ^g^ 273 K, 100 bar; ^h^ 243 K, 100 bar.

In practical applications of MOFs, many non-ideal conditions also need to be taken into account. Firstly, the conditions of raw gas are not only CH_4_ and N_2_: moisture is also typically impure in raw gas. The polar adsorption sites in MOFs are more easily combined with polar H_2_O molecules, resulting in a decrease in adsorption capacity. Therefore, finding MOF materials with excellent anti-moisture stability and low water affinity for CH_4_/N_2_ separation in humid environments remains a challenge. Guo prepared an ultra-microporous Cu-based MOF material, NKMOF-8-Br, which showed a high CH_4_ adsorption capacity (1.84 mmol/g) and a high CH_4_/N_2_ (CH_4_:N_2_ = 50:50) separation selectivity (8.9) at 298 K and 1.0 bar. Profited by the strong coordination bond between Cu(I) and the nitrogen atoms of the ligand, the skeleton of NKMOF-8-Br could remain in boiling water and acidic conditions, and the adsorption performance was not significantly reduced, with excellent structural stability [83]. Then, Li et al. used DMOF as the basic framework, replaced the BDC ligand with 1,4-naphthalene dicarboxylic acid (1,4-NDC, with two aromatic rings) and 9,10-anthracene dicarboxylic acid (ADC, with three aromatic rings), and prepared two functionalized DMOFs (DMOF-N and DMOF-A2). Due to the introduction of more hydrophobic aromatic rings in the structure, DMOF-A2 not only had a higher CH_4_ adsorption capacity (37 cm^3^/g) and CH_4_/N_2_ selectivity (7.2), but also showed a low moisture adsorption capacity, and maintained a high CH_4_/N_2_ separation performance even in an environment with a humidity of 40% [84]. In addition, most of the reported MOFs are powder materials, which may confer high stacking strength when loaded in the adsorption column, which is likely to cause structural cracking in MOFs with low mechanical strength, resulting in a decrease in adsorption capacity. Finding an effective method with which to process the powder MOF material to a size similar to that of commercial carbon adsorbents, ensuring the adsorption capacity at the same time, is very important for the practical application of MOFs. Singh compared the adsorption performances of Cu-BTC particles (B-5, B-10, and B-15) prepared with carboxymethyl cellulose as a binder with the commercial zeolite adsorbents 13X and 5A, and found that although the adsorption capacity on Cu-BTC decreased after extrusion molding, the CH_4_/N_2_ selectivity of B-5 for the CH_4_/N_2_ (CH_4_:N_2_ = 20:80) was still maintained at 3.21, which was about 38% and 34% higher than that of zeolite 13X and 5A, respectively, and B-5 had commercial application value [85]. Moreover, compounding with other materials is also an effective method of solving problems such as structural regulation and stability faced by MOFs. In 2019, a novel method was reported to obtain a flexible MOF with a stable structure, for which spherical mesoporous zeolite SBA-15 was used as the substrate, and nickel-based MOF was grown on the surface to obtain the composite material MOF-2/SBA-15. Due to the optimized structure, the specific surface area and CH_4_ adsorption capacity of MOF-2/SBA-15 were higher than those of the original MOF-2, and the adsorption selectivity parameter S reached 11.1 [86]. Then, Al-Naddaf prepared a core–shell structured 5A zeolite and an MOF-74 hybrid material, Zeo-A@MOF-74. At room temperature and 20 bar, the adsorption capacities of CH_4_ and N_2_ were 7.7 mmol/g and 6.7 mmol/g, respectively, which was about 30% higher than that of MOF-74 [87]. Zheng reported a composite material, Ac-MIL-101, with a micropore size mainly distributed at 1.2 nm obtained by an in situ hydrothermal method to dope carbon nanofibers into MIL-101 crystals. The surface of the crystal was passivated without affecting the crystal structure. The reduced crystal size made the micropore structure more developed, and the adsorption selectivity S for CH_4_/N_2_ mixed gas was greater than 2 [88].

In fact, due to the wide variety of MOFs, the traditional preparation–characterization process did not have a high enough efficiency to screen out MOFs with excellent performance. Introducing big data analysis methods into the construction of MOF structures and the analysis of adsorption and separation properties to guide material design is a very promising direction for the rapid screening and optimization of MOFs in the future. In 2022, Demir combined Grand Canonical Monte Carlo (GCMC) and molecular dynamics (MD) simulations to explore the effects of different functional groups of the CH_4_/N_2_ separation performance on MOFs, and found that MOFs with ligands with two to three functional groups exhibited better CH_4_/N_2_ separation performance, and the -OCH3 group caused the greatest enhancement in CH_4_/N_2_ selectivity [89]. Gulbalkan used high-throughput computing and molecular simulations to study the adsorption and separation performance on 5034 MOF and COF materials for CH_4_/N_2_, and systematically screened the best adsorbent based on adsorption selectivity, working capacity, adsorbent performance score, and renewability. In their work, DFT calculations also found that adding ionic liquids (ILs) to MOFs can significantly improve the CH_4_/N_2_ separation selectivity, at the expense of a partial loss in adsorption capacity [90]. At present, the database of several MOFs has been built. And as research deepens, more MOFs will be added to the database in the future, showing broad prospects.

In summary, research into CH_4_/N_2_ separation on MOFs has made great progress. With a large specific surface area, high porosity, and flexible pore structure, MOFs have good application potential in CH_4_/N_2_ separation, and MOFs with a CH_4_/N_2_ separation selectivity of more than 20 have been synthesized. However, there are still some problems in realizing the industrial application of MOFs. First of all, preparing high-performance MOFs efficiently and at low cost is a big challenge. Although the structure of MOFs can be precisely regulated under laboratory conditions to obtain high adsorption capacity and selectivity, it is difficult to precisely control the structure and performances of MOFs when the preparation of MOFs is scaled up to large-scale and industrial production. The high preparation cost and low output efficiency make it difficult to give full play to the excellent adsorption performances of MOFs. Due to the structural characteristics of the combination of metal ions and organic molecular ligands, capillary collapse is prone to occur in harsh environments (such as acidic, alkaline, and high humidity conditions), destroying the crystal structure and failing to ensure structural stability. Secondly, MOFs have a strong adsorption capacity for certain impure gases, and the toxic effects of impurities will seriously affect the service life of MOFs. In addition, most MOFs in studies are powder materials. In practical applications, the adhesives used in the granulation step will break crystal structures, and the adsorption capacity of MOFs will be significantly reduced when the packing density is high in the adsorption column. In addition, although a database has been established to screen out MOFs with excellent performance, there is still a big margin for improvement in the database.

### 3.3. Porous Carbons

Porous carbons are a type of traditional porous material that are widely used in several fields, such as air separation, sewage purification, catalysis, and energy storage. Porous carbons have the advantages of good structure strength, acid and alkali resistance, hydrophobicity, good regeneration performance, and low cost, and have wide application prospects in CH_4_/N_2_ separation. According to differences in pore structures, porous carbons can be divided into activated carbons (ACs) and carbon molecular sieves (CMSs). The pore size distribution of AC is wider, and the abundant surface functional groups make CH_4_ preferentially adsorbed on AC. The pore size of CMS is more uniform, with a narrower pore size distribution usually concentrated at 0.3–1.0 nm, and makes N_2_ diffuse in pores preferentially, while CH_4_ is blocked outside the pores for a relative short period of time.

#### 3.3.1. Activated Carbons

As a typical porous carbon material, activated carbons usually have a developed pore structure, large pore volume, and high specific surface area that exhibit large adsorption capacity: the characteristics of excellent structural stability and easy desorption and regeneration make activated carbons suitable for gas separation. The structure of activated carbon is formed by the disordered stacking of single or multi-layer graphite, and the cross-linked graphite micro-crystals form porous structures which provide a large space for gas adsorption. In addition, the pore sizes of activated carbons are disordered with wide pore size distributions, containing ultra-micropores, micropores, mesopores, and macropores. Compared to most adsorbents, the surface of activated carbon is non-polar or weak-polar, which makes activated carbons the only adsorbent that does not need to be strictly dried before adsorption, and which can adsorb more non-polar or weak-polar gas molecules. At present, a variety of commercial activated carbons have been developed for CH_4_/N_2_ separation. These activated carbons have different adsorption capacities for CH_4_ due to their different pore structures [6,91,92]. Moreover, many activated carbon adsorbents with great adsorption performances have been reported, as shown in Table 4.

Coal has been widely used to prepare activated carbons for CH_4_/N_2_ separation with the equilibrium effect due to its advantages of low cost and high carbon content. Gao used pitch and coal powder as raw materials to prepare an activated carbon disk (ACD) via potassium hydrate activation. The ACD not only had abundant multi-level pore structures and a large specific surface area, but also exhibited a high mechanical strength (4.3 MPa). The ACD with improved foaming performance by nitric acid pitch treatment showed high CH_4_ adsorption capacity, which was up to 5.48 mol/kg at 298 K and 3500 kPa [93]. Zheng synthesized the activated carbons DF-AC and SM-AC from low-rank bituminous coal and activation with potassium hydrate, and studied the effect of temperature for CH_4_/N_2_ separation performance. It was found that the separation factors increased rapidly with the increase in temperature, and then decreased slowly, while the adsorption selectivity dropped sharply at low pressures and then showed a slow downward trend. The results showed that lower temperatures are conducive to enriching CH_4_ from low-concentration coalbed methane, and the effect of temperature on adsorption is greater than that of pressure [94]. Then, Toprak used bituminous coal as a raw material and prepared activated carbon AC with a large specific surface area (1881 m^2^/g) and pore volume (1.12 cm^3^/g) by using phosphoric acid activation improved by ultrasonic treatment. At 298 K and 1 bar, the CH_4_ adsorption capacity of AC was 19.45 cm^3^/g. The results showed that CH_4_ adsorption mainly occurred in micropores of less than 10 Å, and the presence of oxygen-containing groups on the surface increased CH_4_ adsorption capacity [95]. After that, a series of ACs were prepared from Guhanshan coal with potassium hydrate activation, and the effect of pre-oxidize with hydrogen peroxide was studied. It was found that pre-oxidation could increase the surface O content by 44.6%. Although the adsorption strength of CH_4_ on the activated carbon surface was reduced by an average of 57%, the CH_4_ adsorption capacity was significantly increased. At 293 K and 60 bar, the adsorption capacity could reach 14 mmol/g [96]. Then, Hamyali reported high-performance activated carbon with a large specific surface area (4012 m^2^/g) and high pore volume (2.07 cm^3^/g), which was prepared by controlling the activation conditions. The CH_4_ adsorption capacity on activated carbon at 298 K and 50 bar was as high as 12 mmol/g. Density functional theory (DFT) was used to analyze the binding energy and adsorption mechanism, and it was found that the position of the defective carbon atoms is the most effective position for CH_4_ adsorption, and a high content of defective carbon atoms is conducive to CH_4_/N_2_ separation [97]. Afterwards, coal-based activated carbons C-TCs were prepared by a novel hydrothermal activation method. The results showed that hydrothermal activation with KOH solution could effectively improve the pore structure of activated carbons, significantly increase the pore volume in the range of 0.5–1.0 nm, and form carbon nanotubes as channels for gas transportation, as shown in Figure 8. At 298 K and 101 kPa, the adsorption capacity of CH_4_ on C-TC1.25 was as high as 1.72 mmol/g, and the CH_4_/N_2_ selectivity was 5.5 [98].

Polymer materials are another kind of precursor with unique advantages used to prepare activated carbons. First, when preparing polymer precursors, different types of monomers can be flexibly selected based on different requirements. Especially when specific elements need to be contained, selecting monomers containing the elements is a very effective strategy. And the low content of impurities in polymer materials is conducive to activation and pore formation. In addition, the composition and structural characteristics of polymer materials ensure the uniformity of the pore structure in activated carbons, which provides space for gas adsorption, and the uniform pores are useful for the separation of specific gas molecules. Du reported a template method used to synthesize a novel microporous carbon material, PGC-x. During the polymerization of pyrogallol and glyoxylic acid, the zinc compound template was confined in situ in the polymer framework. Then, thermal treatment was performed to obtain ultra-microporous structures with pore sizes in the range of 0.48–0.57 nm, effectively eliminating mesopores and macropores. At 298 K and 1.0 bar, PGC-1 exhibited the largest CH_4_ adsorption capacity (1.50 mmol/g), and the CH_4_/N_2_ separation selectivity (CH_4_:N_2_ = 15:85) reached 5.8 [99]. Che prepared MgO-doped polyacrylonitrile fibers by electrospinning technology, and then activated carbon nanofibers, ACNFs, were obtained by activation. The addition of MgO reduced the diameter of the nanofibers to 520 nm, and the specific surface area increased by about four times to 2893 m^2^/g, and the ACNFs exhibited a high CH_4_ adsorption capacity (2.37 mmol/g) [100]. Then, carbon microspheres, PCMs, with pore sizes of 0.3–0.6 nm were synthesized by carbonizing a polymer of poly (cyclophosphazene-co-4,4′-sulfonyldiphenol (PCS)) at high temperatures. At 298 K and 100 kPa, PCMs showed excellent CH_4_ adsorption capacity (42.22 cm^3^/g) and high CH_4_/N_2_ selectivity (4.64), which were attributed to their excellent pore structures [101].

Biomass is also an excellent precursor with a high carbon content with which to prepare activated carbons. The structures of biomass usually have a certain mechanical strength, which can ensure the integrity of the pore structure during activation and avoid the collapse of the three-dimensional skeleton. In addition, biomass has obvious advantages in the preparation of heteroatom-doped activated carbons because of its increased heteroatom components, and heteroatom-doped activated carbon can be obtained by a simple activation process. Qu studied the effect of activation conditions on the adsorption and separation performances of coconut shell-based activated carbons, GACs. It was found that KOH activation could effectively reduce the polar functional groups on the surface and enhance CH_4_ adsorption capacity, and CH_4_ was concentrated from 21% to 50% in the four-step single-bed VPSA process [102]. Then, two kinds of nitrogen-doped biomass-activated carbons (BACs and GACs) derived from banana peel and grapefruit peel were synthesized for CH_4_/N_2_ separation, and the preparation method is shown in Figure 9. The results showed that the high surface nitrogen content could effectively improve the CH_4_/N_2_ separation ability, and CH_4_/N_2_(30/70) binary mixture adsorption selectivity over BC600 reached 3.98 at 25 °C, and GC600 reached 5.84 at 0 °C [103]. Liu prepared N and O-rich activated carbons by NaNH_2_ activation from palm sheaths at 350–550 °C, which showed a large specific surface area (2181 m^2^/g), a narrow pore size distribution, and abundant heteroatom groups. It was found that the synergistic effect of high ultra-micropore content and an appropriate polar group ratio resulted in excellent gas separation performance, with a CH_4_/N_2_ selectivity of 7.6 at 298 K and 1.0 bar [104]. Li reported a N-doped porous carbon N-WAPC derived from KOH-activated waste wool and urea treatment. N-WAPC exhibited a high N content (14.48%), narrow micropores (0.52–1.0 nm), and a high specific surface area (862 m^2^/g). At 298 K and 1.0 bar, the adsorption amounts of CH_4_ and N_2_ were 1.01 mmol/g and 0.13 mmol/g, respectively, and the separation selectivity predicted by IAST was 7.62 [105]. Tang prepared binder-free granular activated carbons (PRCs) from rice, with a specific surface area of 776 cm^3^/g. At 298 K and 100 kPa, the CH_4_ adsorption capacity reached 1.12 mmol/g, and the CH_4_/N_2_ separation ratio was 5.7, which showed good separation ability. In addition, the contribution of hydroxyl, carboxyl, and aldehyde groups to the CH_4_/N_2_ separation ability was explored through computational simulation, and it was found that the carboxyl group played an important role in CH_4_ adsorption [106]. In 2021, carbon microspheres AC-KP were synthesized from glucose by hydrothermal treatment with a mixed solution of potassium permanganate and potassium hydroxide. AC-KP exhibited uniform pore size distribution (of mainly 0.4–0.7 nm) and high ultra-micropore volume (0.50 cm^3^/g). The CH_4_ adsorption capacity was 1.87 mmol/g, and CH_4_/N_2_ selectivity was 6.5 at 298 K and 101 kPa, which showed excellent separation performances [107].

Current experiments and simulations have proved that the adsorption of CH_4_ and N_2_ mainly occurs in micropores on activated carbons, and the structure of ultra-micropores plays an important role in separation selectivity. Therefore, a lot of studies have focused on analyzing the influence of pore structure on CH_4_/N_2_ separation, and guide the preparation of high-performance activated carbons. Yuan et al. investigated the adsorption behavior of CH_4_ and N_2_ on a series of activated carbons (CACs) with similar surface properties but different pore structures, and systematically analyzed the relationship between micropore structures and adsorption performances. The experimental results showed that the adsorption capacity was positively correlated with the capacity of micropores, and a micropore content in the range of 0.55–0.85 nm had the greatest impact on the CH_4_/N_2_ equilibrium selectivity, while the mesopores and macropores had little effect on the selectivity [108]. Then, Xu systematically analyzed the relationship between the separation performances and pore structure parameters of coal-based activated carbons. It was found that the effective pores for separating CH_4_/N_2_ were pores smaller than 1 nm, and the optimal pore size range was 0.4–0.7 nm. the concept of “effective pore percentage” was also proposed, which was the proportion of pores in the pore size range of 0.4–0.7 nm in pores smaller than 1 nm. The results of ridge regression analysis showed that the higher the effective pore percentage, the more obvious the improvement in separation efficiency, while a high proportion of ineffective pores reduced the separation efficiency. Therefore, it is necessary to avoid blindly increasing the specific surface area and pore volume of activated carbons, and increasing the pore volume with a high effective pore percentage can improve the overall separation performances of activated carbons [109]. Zhang studied the adsorption performances of anthracite-based activated carbons (GACs) with similar surface chemical properties but different pore structures. The results showed that although the specific surface area and pore volume of the micropores played an important role in the separation performance, the pore size distribution was the determining factor, and the main pore size range affecting the adsorption was 5–10 Å. The optimal GAC had a CH_4_ adsorption capacity of 2.54 mmol/g at 298 K and 2.0 MPa, and it also had a CH_4_/N_2_ adsorption selectivity of 3.23. And in a single-column PSA experiment, the CH_4_ concentration increased from 30% to 63.5% [110]. Then, Kiełbasa used KOH and K_2_CO_3_ to activate with commercial activated carbons, FPV, WG-12, and CWZ-22, to enhance their pore volume. It was found that the adsorption amount of CH_4_ on the modified activated carbons was strictly related to the microporosity or specific surface area, and the modified FPV-activated carbon showed the best performance, with the largest CH_4_ adsorption capacity (9.7 mmol/g) at 298 K and 35 bar [111].

In addition to the pore structures, the surface properties also have a significant impact on the CH_4_/N_2_ separation ability of activated carbons. Increasing the content of specific surface functional groups or introducing specific heteroatoms can effectively increase the CH_4_ adsorption capacity, thereby enhancing the performance of activated carbons. Heteroatom-doped activated carbons can be obtained from the carbonization of precursors containing specific elements or from the doping and modification on activated carbon precursors. In 1990, Baksh tried to deposit molybdenum dioxide onto activated carbon Super A to improve the CH_4_/N_2_ separation selectivity. It is found that the CH_4_ adsorption capacity and specific surface area of doped activated carbons decreased with the increase of the doping amount, but the equilibrium separation selectivity increased significantly. The activated carbon with 12 wt.% molybdenum dioxide showed the highest CH_4_/N_2_ separation selectivity (4.25), and the CH_4_ concentration could be increased to 90% with a recovery rate of 73% in the PSA experiment. Feng modified the surface properties of carbon microspheres with urea, ammonia, and NH_3_ treatment, respectively, and found that the differences in pore structures could not explain the adsorption behavior of CH_4_, and CH_4_ adsorption depended on the polarizability of adsorbents, which effected the van der Waals forces between adsorbents and adsorbates. With similar pore structures, the carbon microsphere with a higher nitrogen content and high N/C ratio showed stronger CH_4_ adsorption capacity, and especially higher pyridinic nitrogen, which provided more adsorption sites for CH_4_ adsorption [112]. And then, activated carbons were modified with different solutions. The CH_4_ adsorption mainly occurred in micropores, and the surface alkaline groups could enhance the CH_4_ adsorption capacity. Benefited by the high micropore volume and the high content of alkaline groups, KCl/AC exhibited the largest CH_4_ adsorption capacity. At 298 K and 1 atm, the CH_4_ adsorption capacity on KCl/AC was 7.89 cm^3^/g, which was 38.9% higher than that of the original AC (5.68 cm^3^/g). The CH_4_/N_2_ separation selectivity was 5.33, which was 38.4% higher than that of AC (3.85). Yang prepared activated carbon SACs through KOH activation from semi-coke, and investigated the effects of modification with organic reagents (benzene, methyl benzoate, and ethyl acetate) and low-temperature plasma treatment (CH_4_ and N_2_) on adsorption and separation performances. It was found that the modification with organic reagents could introduce functional groups on SACs, while low-temperature plasma treatment could further enhance the content of these functional groups. SAC-ben-P-N_2_ exhibited the highest CH_4_/N_2_ separation selectivity (5.08), and the breakthrough time of CH_4_ was 93 s. In addition, the breakthrough time of CH_4_ on SAC-ben was the longest (112 s), and the CH_4_/N_2_ separation selectivity reached 4.87 [113]. Li constructed a N-doped amorphous carbon model by way of the Monte Carlo method, and the neural network was trained to predict the adsorption capacities of CH_4_ and N_2_ based on factors of nitrogen content, specific surface area, pore size, and atomic charges. It was revealed that the selectivity was significantly enhanced by nitrogen content in micropores smaller than 5 Å, and the interaction between doped nitrogen and non-polar C-H bonds played a major role in CH_4_ adsorption [114]. Then, Ni-doped porous carbons were built using the DFT method, where Ni was incorporated to control the pore structure and surface properties. It was found that doped Ni created a synergistic effect between the optimization of pore size and the change in adsorption potential, leading to an increase in CH_4_ adsorption capacity (Figure 10). The porous carbon with 10% Ni/C exhibited the highest CH_4_ adsorption capacity (28.3 cm^3^/g) and the highest CH_4_/N_2_ separation selectivity (5.29) at 298 K and 1 bar [115]. Subsequently, Zhang reported a novel type of sulfur-doped activated carbon microsphere, where sulfur was incorporated through hydrothermal treatment with glucose solution and sodium dodecylbenzenesulfonate (SDS). As the sulfur content increased, the methane (CH_4_) adsorption performance of the carbon microspheres gradually improved. At 273.15 K and 101 kPa, ACS-700-2 exhibited the optimal performance, demonstrating not only a high CH_4_ adsorption capacity (2.98 mmol/g), but also an exceptionally high CH_4_/N_2_ separation selectivity (22.8), higher than most reported activated carbons [116]. Recently, Mujmule reported glucose-based activated carbon HTACs with a high specific surface area exceeding 2000 m^2^/g via the hydrothermal treatment of a mixed solution containing urea and glucose, and the schematic illustration of the synthesis of porous carbon adsorbents is shown in Figure 11. It was observed that the structural properties of adsorbents, including specific surface area and polarity, could be significantly influenced by adjusting the amount of urea addition, thus the adsorption capacity was enhanced. At 298.15 K and 100 kPa, 1.5-HTAC exhibited the highest CH_4_ adsorption capacity of 36.86 cm^3^/g and a CH_4_/N_2_ separation selectivity of 6.29 [117].

Overall, activated carbon is considered a potential high-efficiency adsorbent material for CH_4_/N_2_ separation, owing to the advantages of large specific surface area, high pore volume, low cost, and excellent structural stability. However, the contradiction between CH_4_ adsorption capacity and CH_4_/N_2_ separation selectivity greatly limits CH_4_/N_2_ separation on activated carbons. Currently, by rationally controlling the pore structures and surface properties of activated carbons through the selection of raw materials, the optimization of activation conditions, and appropriate post-treatment methods, the adsorption capacity of CH_4_ on activated carbons can be effectively enhanced, thereby significantly improving CH_4_/N_2_ separation performances. Notably, the incorporation of heteroatoms such as N and O into activated carbons to modify the thermodynamic adsorption strength between CH_4_ and activated carbons has become an effective method of enhancing CH_4_ adsorption capacity on activated carbons. However, the pore structures of activated carbons are formed by the stacking of amorphous graphite microchips, which make the controlled doping heteroatoms challenging. At present, only the overall content of doped heteroatoms can be modified, and the concentration of heteroatoms within specific pore size ranges and precise locations in pores cannot be accurately controlled, making it difficult to achieve the optimal doping state. In conclusion, the development of activated carbons of high performances remains a key challenge for effective CH_4_ capture and CH_4_/N_2_ separation in the future. It is imperative to develop methods of precisely controlling pore size distribution, tuning surface chemical properties, and reducing production costs.

#### 3.3.2. Carbon Molecular Sieves

A carbon molecular sieve (CMS) is a type of porous carbon with uniform pore sizes. The pore structure of a carbon molecular sieve is predominantly composed of micropores, with a minor presence of mesopores and macropores serving as pathways for gas transport. CH_4_/N_2_ separation on CMS is mainly based on the kinetic effect, which is directly influenced by their pore size. The kinetic effect predominates when the pore size is between the kinetic diameters of CH_4_ and N_2_, and the separation efficiency is significantly enhanced by the difference in diffusion rates of CH_4_ and N_2_ in the pores. Relatively speaking, the preparation of high-performance CMS for kinetic CH_4_/N_2_ separation is a great challenge. On the one hand, due to the very small difference in the kinetic diameters of CH_4_ and N_2_, it is difficult to obtain CMSs with the appropriate pore size. On the other hand, although the diffusion rate of CH_4_ in pores is lower than that of N_2_, CH_4_ is preferentially adsorbed in pores because of its higher polarizability. This results in a conflict between the thermodynamic effect and the kinetic effect for CH_4_/N_2_ separation, weakening the kinetic separation efficiency. In early research, CMSs were primarily synthesized by the carbonization of polymers (such as PVC and Saran resin), cellulose, and coconut shells. Additionally, CMSs could also be produced by the mild carbonization of coals with high carbon content and high volatile matter [118]. Although the pore size distribution is relatively uniform, low specific surface area and low pore volume are also exhibited on these CMSs, thus the adsorption capacity is low. Appropriate activation is an effective method of increasing the specific surface area and pore volume of CMS. However, activation is a complex process involving multiple reactions that is difficult to be accurately controlled. As a result, CMSs that are only synthesized by carbonization and activation usually exhibit a wide pore size distribution, which is insufficient for kinetic separation, and adjusting the pore size after activation would significantly enhance kinetic separation efficiency. Carbon deposition is a frequently-used method for pore size adjustment, typically using organic reagents such as hydrocarbons, benzenes, or asphaltenes as deposition agents. At a high temperature, these agents pyrolyze into free carbon and deposit in pores. For a given deposition process, it is necessary to comprehensively control conditions such as agent concentration, temperature, and carbon deposition time. Ideally, free carbon should be deposited only at the pore entrances rather than throughout the entire pores to ensure both high adsorption capacity and high kinetic efficiency [119,120,121]. In addition, low temperature plasma deposition can also be used for reducing pore size [122]. Generally, carbon deposition implies that CMSs contain only a low content of surface functional groups, which is different from activated carbons that possess abundant surface functional groups. This results in the CH_4_ and N_2_ adsorption on CMSs primarily relying on dispersion forces between the carbon sheets forming the pores, with small contributions from ion-induced dipole interactions by surface functional groups.

Research on CMSs for gas separation began early and may date back to 1980s, and significant progress has been made. Currently, several commercial CMSs have been developed for the kinetic separation of CH_4_/N_2_. In 1993, Loughlin measured the adsorption parameters of CH_4_ and N_2_ on carbon molecular sieves (CMSs) using chromatography, volumetric, and gravimetric methods, respectively. It was found that the adsorption rate measured by the volumetric method was inconsistent with the diffusion hypothesis, possibly due to the coexistence of barrier resistance and diffusion resistance. However, the diffusion resistance and dual-resistance results obtained by chromatography and gravimetric methods were in agreement [123]. Cavenati analyzed the thermodynamic and kinetic adsorption behaviors of CH_4_ and N_2_ on a commercial carbon molecular sieves (CMSs 3K) at various temperatures by the gravimetric method. The results indicated that the adsorption behaviors of both gases were controlled by a combination of pore entrance resistance and intraparticle diffusion resistance. Although the Linear Driving Force (LDF) model predicted a N_2_/CH_4_ kinetic constant ratio of 133, the overall kinetic selectivity at 308 K was only 1.9 due to the relatively high CH_4_ adsorption capacity [124]. Then, the adsorption thermodynamic equilibrium and kinetics of CH_4_ and N_2_ on three commercial carbon molecular sieves (CMSs) were investigated. It was found that an appropriate average pore size is a crucial factor for enhancing the kinetic separation performance of CMSs. Furthermore, the separation selectivity indicated by the breakthrough curves deviated from the conventional N_2_/CH_4_ kinetic selectivity, which is attributed to the underestimation of the influence of N_2_ equilibrium adsorption on its kinetic adsorption. Therefore, a modified selectivity with consideration of the actual N_2_ adsorption capacity was proposed, showing the correct performance order [125]. In addition, the adsorption isotherms of CH_4_ and N_2_ on a commercial CMS (CMS-131510) at 0–700 kPa and 303–343 K were measured, and the adsorption kinetics of pure gases at different surface coverages were investigated by the batch adsorption experiment. This study revealed that the kinetic selectivity exhibited a dependence on both the temperature and the concentration of the mixed gases, and the influence of micropore distribution on the diffusion rate was discussed [126]. Furthermore, Zhang found that the pore structure of anthracite-based carbon molecular sieves (CMSs) can be significantly altered through the chemical vapor deposition (CVD) of benzene. Due to the optimization of the pores, CMS-2 exhibited a high CH_4_/N_2_ adsorption selectivity (8.62) [127]. Gorska synthesized carbon molecular sieves (CMSs) with a narrow pore size distribution centered at 0.8 nm from willow woods for the separation of various gas mixtures. At temperatures ranging from 30 to 70 °C, the CH_4_/N_2_ separation factor was found to be 10.20 to 3.64. Further experiments also indicated that the gas separation mechanism is based on geometric factors, closely related to the sizes and shapes of gas molecules [128]. Subsequently, Zhang prepared CMS for N_2_/CH_4_ kinetic separation using activated carbon as the precursor through the chemical vapor deposition (CVD) of benzene. It was found that the deposition temperature had a significant impact on the structural modification of CMS. The CMS-G exhibited a high kinetic selectivity (35.26), and the enrichment of CH_4_ reached 30.20% in a single-column pressure swing adsorption (PSA) experiment, showing great kinetic performance [129]. After that, coal-based carbon molecular sieves (CMSs) were modified by affinity organic reagents (tetracosane, sodium dodecyl sulfate, and polyimide) and low-temperature plasmas (CH_4_ and N_2_). It was found that the modified CMSs exhibited enhanced CH_4_ adsorption capacity. At 298 K and 2 MPa, the adsorption capacities of CH_4_ and N_2_ on the CMSs modified with polyimide and N_2_ plasma were 6.76 mmol/g and 5.56 mmol/g, respectively, with a CH_4_/N_2_ separation factor of 3.32 [130]. Then, Liu synthesized carbon molecular sieves via the pyrolysis of a gel-type ion-exchange resin (IER) based on sulfonated poly(styrene-divinylbenzene). Nine different gas probe molecules were employed to determine the critical pore size of the CMSs. The CMS adsorbents exhibited micropore sizes ranging from 3.5 to 4.6 Å and demonstrated a N_2_/CH_4_ separation factor of 9 [131]. Subsequently, Liu further investigated the separation performances of carbon molecular sieve fibers (CMSFs) prepared by the pyrolysis of melt-extruded Polyvinylidene chloride (PVDC). It was found that as the pyrolysis temperature increased, the effective micropore size gradually decreased from 4.9 Å to 3.4 Å, improving adsorption kinetics [132].

**Table 4 nanomaterials-15-00208-t004:** Adsorption performances of activated carbons for CH_4_/N_2._

Activated Carbons	CH_4_/N_2_ Selectivity	CH_4_ Adsorption Capacity (mmol/g)	Reference
ACD	-	5.48 ^a^	[93]
H800N_2_-S	-	0.70 ^d^	[95]
AC-5	-	14 ^b^	[96]
IACA-5-60-900	-	12 ^c^	[97]
C-TC1.25	5.5 ^d^	1.72 ^d^	[98]
PGC-1	5.8 ^d^	1.50 ^d^	[99]
ACNF MgO	-	2.37 ^e^	[100]
PCM	4.64 ^d^	1.88 ^d^	[101]
BC600	3.98 ^d^	1.3 ^d^	[103]
GC600	5.84 ^e^	2.1 ^e^	[103]
N WPAC	7.62 ^d^	1.01 ^d^	[105]
AC-KP	6.5 ^d^	1.87 ^d^	[107]
GAC	3.23 ^f^	2.54 ^g^	[110]
10%Ni/C	5.29 ^d^	1.26 ^d^	[115]
ACS-700-2	22.8 ^e^	2.98 ^e^	[116]
1.5-HTAC	6.29 ^d^	1.65 ^d^	[117]
KCl/AC	5.33 ^g^	0.35 ^g^	[133]

^a^ 298 K, 35 bar; ^b^ 293 K, 60 bar; ^c^ 298 K, 50 bar; ^d^ 298 K, 1 bar; ^e^ 273 K, 1 bar; ^f^ 298 K, 20 bar; ^g^ 293 K, 1 bar.

In summary, carbon molecular sieves with uniform pore sizes represent potential adsorbents for N_2_/CH_4_ kinetic separation. The performances of kinetic separation can be enhanced primarily by adjusting the pore size. However, despite the fact that carbon deposition can improve the pore structure and effectively enhance its kinetic separation performance, precise regulation of specific pore size remains highly challenging due to the complexity and disorder of the pore structures. Furthermore, the carbon sources used for carbon deposition, such as benzene and gaseous alkanes, also have certain limitations. For instance, benzene is a toxic chemical compound, which does not align with the trend of green chemistry. Although gaseous alkanes are relatively inexpensive, they also have potential safety hazards. At present, a carbon deposition agent that is simultaneously low-cost and high-safety has not yet been developed. In the future, the development of convenient, efficient, and green methods of pore regulation will significantly promote the application of carbon molecular sieves in CH_4_/N_2_ separation.

## 4. Conclusions

The development of new technologies for CH_4_ capture from coalbed methane with low CH_4_ concentration can not only achieve the exploration and efficient utilization of unconventional natural gas, promoting the green transformation of the energy consumption structure, but can also effectively reduce CH_4_ emissions, contributing to the mitigation of global warming. However, the efficient separation of CH_4_ and N_2_ presents a considerable challenge. At present, various technologies for CH_4_/N_2_ separation have been developed, with cryogenic distillation and adsorption separation being the most notably industrialized. Among these, adsorption separation emerges as a technology with great potential for removing N_2_ from low-concentration CH_4_, owing to its operational flexibility, simple process, and low energy consumption. The key to advancing adsorption separation technology is the development of high-performance adsorbents. Zeolites, metal–organic frameworks (MOFs), and porous carbons have been widely studied as potential adsorbents for CH_4_/N_2_ separation, and a profound understanding of the separation mechanisms and optimization strategies for these adsorbents has been gradually established. Nevertheless, from a material design and development perspective, several critical issues are still unresolved that would allow us to enhance the differential adsorption behaviors of CH_4_ and N_2_ on adsorbents. For instance, the high polarizability of zeolite surfaces poses a challenge, as it prioritizes the adsorption of polar molecules, reducing the overall adsorption capacity for CH_4_. Developing an effective strategy to eliminate the mutual inhibition effects between the kinetic and thermodynamic separation for CH_4_ and N_2_ on zeolite is crucial. MOF materials, with their flexible structural controllability and surface adjustability, offer promising sights for optimizing CH_4_/N_2_ separation performance. However, existing MOFs still face challenges for improvement in terms of enhancing structural stability and reducing production costs. In addition, the pore structure of traditional porous carbon materials cannot be precisely controlled, and methods for the stable and precise doping of heteroatoms within specific pores require further research. Moreover, the preparation costs of these materials need to be reduced to make them more economically viable for industrial applications. In addition, the development of adsorbents lies in matching the properties of adsorbents and adsorbates, and controlling pore structures and tuning surface properties at the atomic scale, which will significantly increase the potential of adsorbents for CH_4_ capture and separation from CBM.

## Figures and Tables

**Figure 1 nanomaterials-15-00208-f001:**
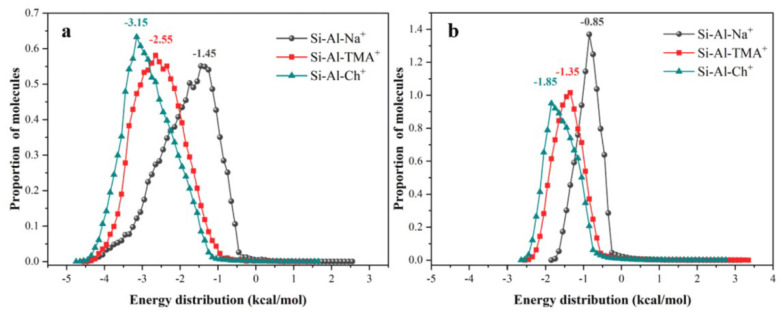
Energy distribution of (**a**) CH_4_ and (**b**) N_2_ during adsorption in Si(1)Al(1) clusters [46].

**Figure 2 nanomaterials-15-00208-f002:**
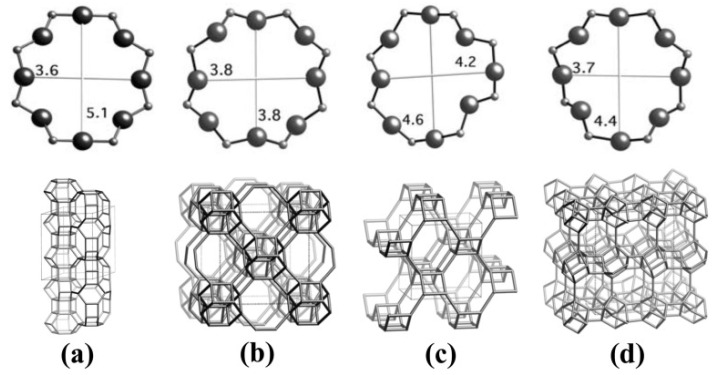
Structures of (**a**) AlPO_4_−17 (ERI), (**b**) AlPO_4_−18 (AEI), (**c**) AlPO_4_−33 (ATT), and (**d**) UiO−7 (ZON) [47].

**Figure 3 nanomaterials-15-00208-f003:**
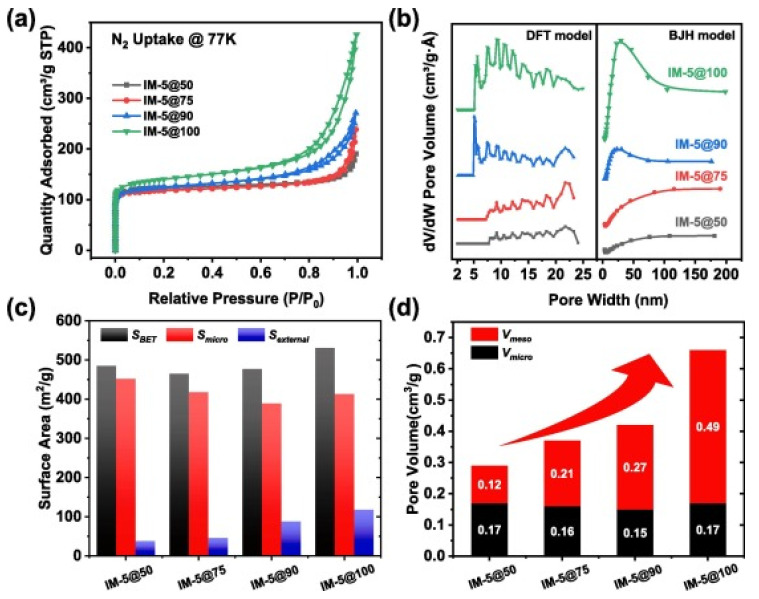
Adsorption and desorption isotherms of N_2_ at 77 K (**a**); mesopore distribution (DFT model in the left region; BJH model in the right region. Both are based on adsorption branching curves) (**b**); histogram of specific surface area (**c**); and histogram of pore volume (**d**) [55].

**Figure 4 nanomaterials-15-00208-f004:**
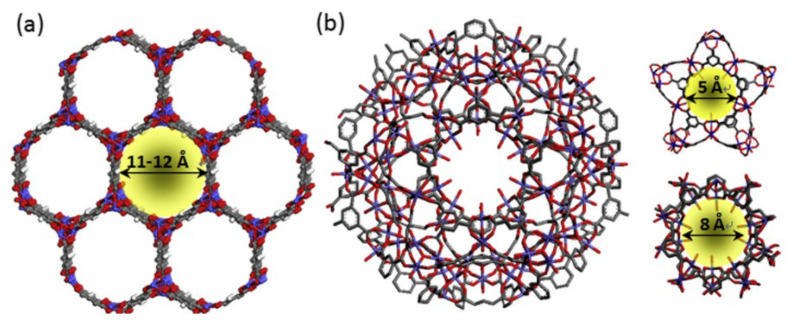
View of the unit cell of the M/DOBDC (Mg, Co, or Ni) structure (**a**) and the MIL−100(Cr) structure (**b**) (M, blue; O, red; C, gray; H, white) [66].

**Figure 5 nanomaterials-15-00208-f005:**
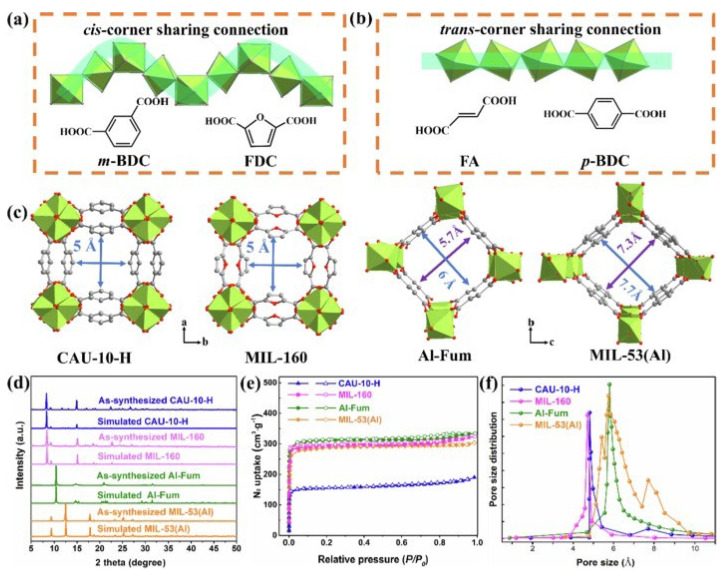
The self-assembly processes through (**a**) helical chains of AlO_6_ polyhedra with m-BDC and FDC linkers; (**b**) linear chains of AlO_6_ polyhedra with FA and p-BDC linkers. (**c**) Square-like and rhombic-like 1D channels in Al-MOFs viewed along the c axis and a axis, respectively. (**d**) The PXRD patterns of Al-MOFs; (**e**) N_2_ adsorption–desorption isotherms of Al-MOFs at 77 K (filled and open symbols represent adsorption and desorption, respectively). (**f**) Pore size distribution of Al-MOFs calculated by HK method [69].

**Figure 6 nanomaterials-15-00208-f006:**
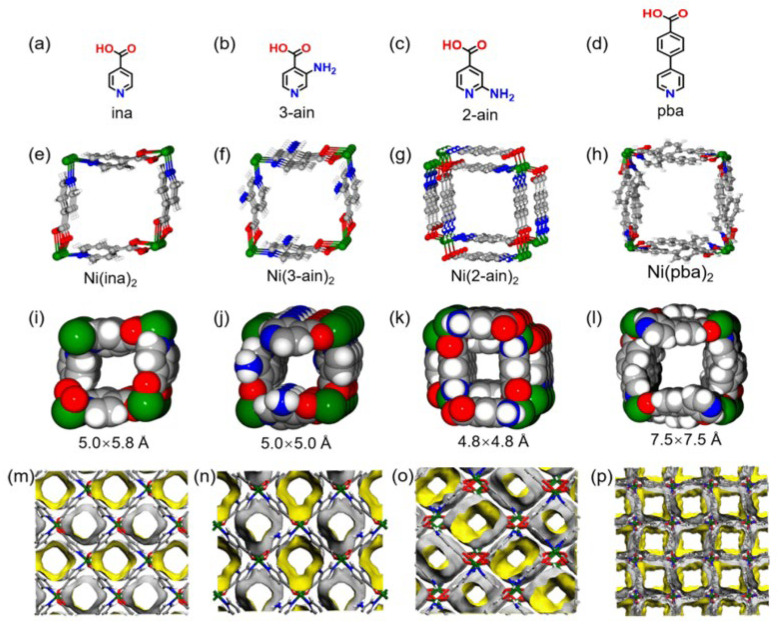
(**a**–**d**) The ligands used for the construction of Ni-MOFs. (**e**–**l**) Pore aperture and pore chemistry for the family of porous materials. Comparison of guest accessible channels of (**m**) Ni(ina)_2_, (**n**) Ni(3-ain)_2_, (**o**) Ni(2-ain)_2_, and (**p**) Ni(pba)_2_ [77].

**Figure 7 nanomaterials-15-00208-f007:**
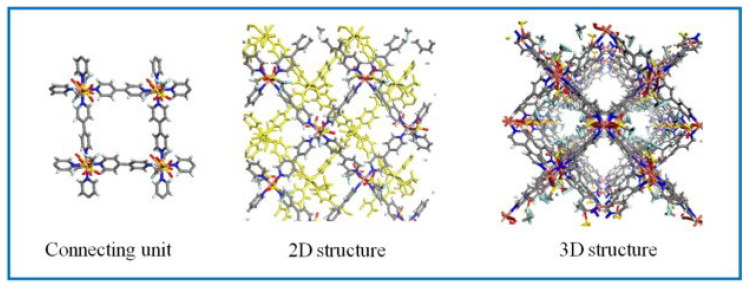
A view of the 2D and 3D structures of the Cu(4,4′-bpy)_2_(OTf)_2_ frameworks [81].

**Figure 8 nanomaterials-15-00208-f008:**
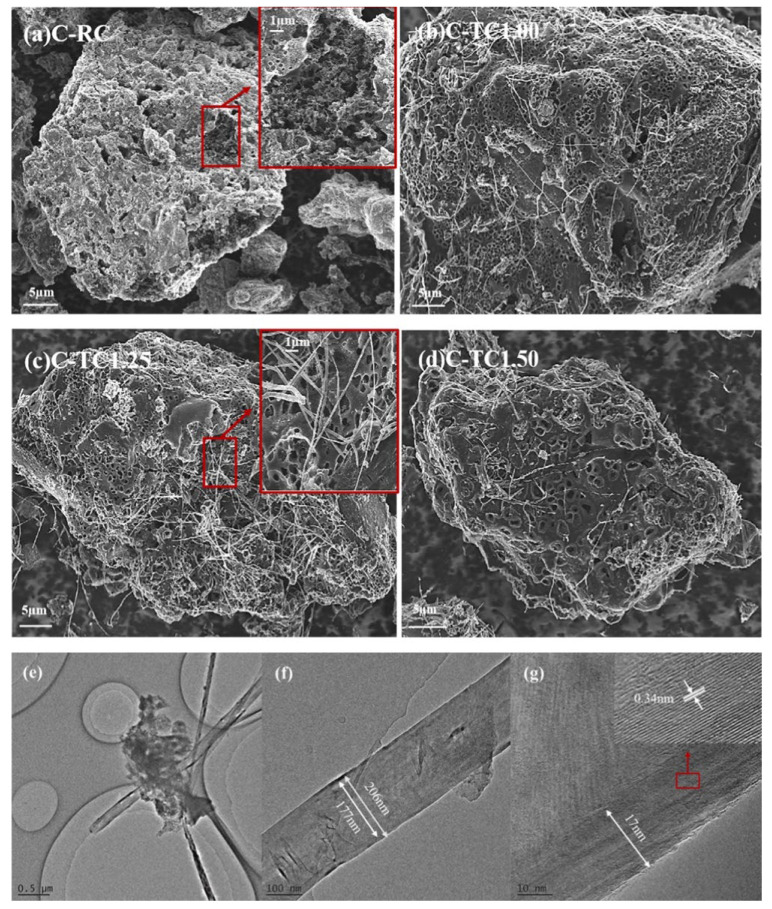
SEM images of coal-based porous carbons: (**a**) C-RC, (**b**) C-TC1.00, (**c**) C-TC1.25, (**d**) C-TC1.50, and (**e**–**g**) TEM images of CNTs in C-TC1.25 [98].

**Figure 9 nanomaterials-15-00208-f009:**
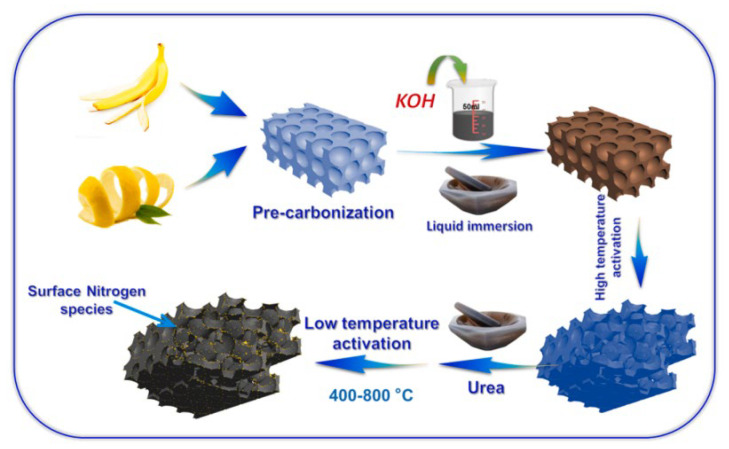
Schematic illustration for fabricating nitrogen-doped activated carbon adsorbent from waste banana peels and grapefruit peels [103].

**Figure 10 nanomaterials-15-00208-f010:**
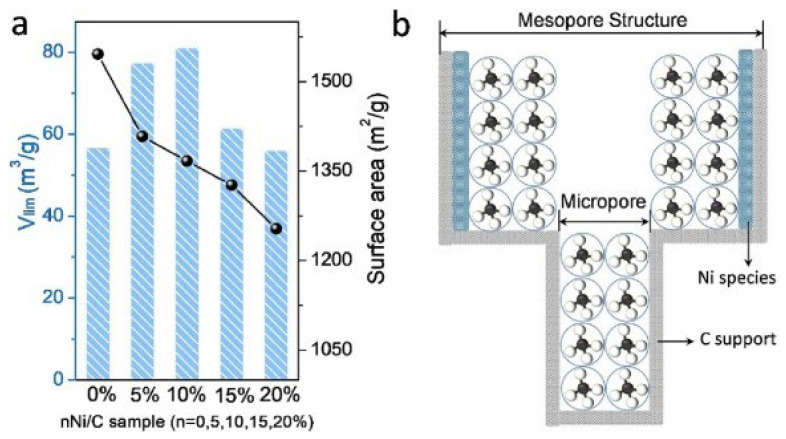
The relationship between the limit adsorption of CH_4_ and pore structure parameters for nNi/C series adsorbents (**a**); schematic diagram of Ni-decorated porous carbon composites enhancing CH_4_ adsorption (**b**) [115].

**Figure 11 nanomaterials-15-00208-f011:**
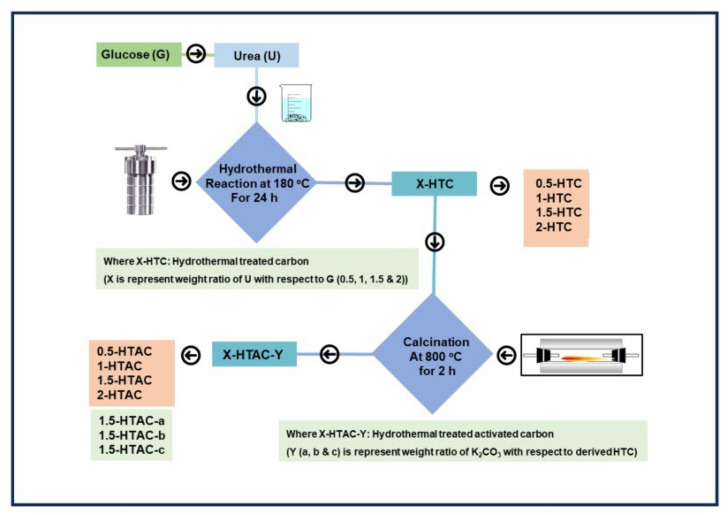
Schematic illustration of the synthesis of porous carbon adsorbents [117].

**Table 1 nanomaterials-15-00208-t001:** Comparison of chemical and physical properties of CH_4_ and N_2._

Molecules	δ/nm	α/cm^3^	μ/(C·m)	T_c_/K
CH_4_	0.380	26.6 × 10^−25^	0	190
N_2_	0.364	17.6 × 10^−25^	0	126

δ: kinetic diameter; α: polarizability; μ: dipole moment; T_c_: critical temperature.

## Data Availability

No new data were created or analyzed in this study. Data sharing is not applicable to this article.

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
