# Peer review of "Microporous Adsorbents for CH4 Capture and Separation from Coalbed Methane with Low CH4 Concentration: Review"

_nanomaterials, 2025, doi:10.3390/nano15030208_

Round 1
Reviewer 1 Report
Comments and Suggestions for Authors
line 73
"the molecular dynamic diameters of CH4 (0.380 nm) and N2 (0.364 nm) are very similar, with a difference of only 0.016 nm."
Doesn't this refer to kinetic diameters?
Writing "very" is in my opinion an over interpretation because this is a typical range for gases. He -0.260 to 0.430 for propane. And e.g. CO - 0.376, O2 - 0.346, C2H4 - 0.390, Ar - 0.340.
In the review article, the authors should take a more critical approach to some reports, such as:
line 680
Toprak [93] publication (common error) has P/Po on the abscissa axes, what is Po?
At temperatures below the critical temperature Po is the saturated vapor pressure. For T>Tc Po has no meaning.
line 688-692
Don't the authors find values of 15-20 mmol/g at room temperatures strange? Assuming porosity of even 2 cm3/g, the density of the collected methane is close to the density at critical conditions.
Similarly, the specific surface area at levels > 2000m2/g?
It would be worth reading the work by Stoeckli and Centeno, doi:10.1016/j.carbon.2004.12.010.
Do the authors think that banana precarbonate (Fig. 5.) is shaped like an egg carton?
What do the authors think about giving values e.g. in [88], (Table 1.) porosity % 64.61+/- 3.14?
Why (Table 3) did the authors not provide values for some references, e.g. [59]?
Author Response
Dear reviewer,
Thank you for your thorough and most helpful comments that have substantially improved the paper. We have revised the paper carefully according to your comments. Details of responses for your comments are in the attachment. Revision sections are shown in red in the revised manuscript.
Kindly regards, Xiao Wei

Reviewer 2 Report
Comments and Suggestions for Authors
This is a very interesting review, well organized and easy to understand. I recommend the publication of this review after answering the following comments.
L 140 For inorganic membranes, the competitive relationship between separation selectivity and permeability makes it hard to separate CH4/N2, because if the selectivity is increased by one order of magnitude, the permeability will decrease by 4-5 orders of magnitude. This is for polymeric membranes not for inorganic membranes. Some examples of the separation using membranes should be added.
L237” N2 preferential diffuses and is adsorbed in pores?. However, in line 232 “CH4 be selectively adsorbed.
L 536 reference number is small
Author Response

(The authors gave the same response as above.)
